# Cardiovascular Fitness and Stride Acceleration in Race-Pace Workouts for the Prediction of Performance in Thoroughbreds

**DOI:** 10.3390/ani14091342

**Published:** 2024-04-29

**Authors:** Charlotte Schrurs, Guillaume Dubois, Emmanuelle Van Erck-Westergren, David S. Gardner

**Affiliations:** 1School of Veterinary Medicine & Science, University of Nottingham, Sutton Bonington, Loughborough LE12 5RD, UK; 2Arioneo, Rue Claude Farrère, 6, 75016 Paris, France; guillaume@arioneo.com; 3Equine Sports Medicine Practice, 83 Avenue Beau Séjour, 1410 Waterloo, Belgium; evanerck@esmp.be

**Keywords:** horse, exercise, training, heart rate, speed

## Abstract

**Simple Summary:**

Data collected from training sessions has the potential to better understand how well racehorses will perform on race-days. To date, no research has explored whether speed during and heart rate recovery after fast exercise could predict a horse winning a race. This theory was tested by analysing information from fitness trackers worn by 485 racehorses during training sessions in Australia by looking at factors like the horse’s speed and heart rate recovery after exercise. The study found that certain factors, like being a colt or a ‘stayer’, were good indicators of future success on the racetrack. However, heart rate recovery and speed at the start or throughout the final 600 m during training were not as powerful as race predictors. Nevertheless, combining physiological data with other less tangible factors will help professionals identify which horses are more likely to perform at their best in races, aiding decision-making and potentially reducing race-day poor performance. Ultimately, this could lead to more successful outcomes for the horses, trainers and owners.

**Abstract:**

In-training racehorse physiological data can be leveraged to further explore race-day performance prediction. To date, no large retrospective, observational study has analysed whether in-training speed and heart rate recovery can predict racehorse success. Speed (categorised as ‘slow’ to ‘fast’ according to the time taken to cover the last 600 m from a virtual finish line) and heart rate recovery (from gallop to 1 min after exercise) of flat racehorses (n = 485) of varying age, sex and type according to distance (e.g., sprinter, miler and stayer) were obtained using a fitness tracker from a single racing yard in Australia. Race-pace training sessions on turf comprised ‘fast gallop’ (n = 3418 sessions) or ‘jumpout’ (n = 1419). A posteriori racing information (n = 3810 races) for all 485 racehorses was extracted and combined with training data. Race performance was categorised as win/not-win or podium or not, each analysed by logistic regression. Colts (*p* < 0.001), stayers (*p* < 0.001) and being relatively fast over the last 600 m of a benchmark test in training (*p* < 0.008) were all predictive of race performance. Heart rate recovery after exercise (*p* = 0.21) and speed recorded at 600 m of a 1 km benchmark test in training (*p* = 0.94) were not predictive. In-training physiological data analytics used along with subjective experience may help trainers identify promising horses and improve decision-making.

## 1. Introduction

The value of any racehorse is highly influenced by how many races it wins. Race-day performance is complex and involves multiple variables; horses compete over different distances (1000 to 2400 m), at different ages (often starting at two years of age), on different surfaces (dirt, turf or synthetic), track layouts (circular versus open) and in varying weather conditions (dry, fast tracks versus wet and heavy tracks). Any trainer’s short- or long-term decision-making for a given racehorse will be influenced by performance in training, performance in early races and other external factors such as the racehorse owners [1].

Across the equine racing world, trainers share the common objective of developing racehorses to perform well across their racing careers. Nevertheless, the trainer’s approach to race-day preparation remains a subjective and singular practice. Some may only rely on experience, while others also incorporate data-driven technology. In Australia, racehorse training typically comprises routine canters (<13.3 m/s), gallops (13.3–14.3 m/s) and pre-trial gallops (≥13.3 m/s), coupled to alternative methods, including mechanical walkers, treadmills and swimming [2]. Barrier trials or ‘jumpouts’ simulate race-day experiences by allowing horses to race at top speed against each other. These sessions are therefore of great interest to the trainer in terms of horse readiness-to-race and accumulated fitness. In a prior study performed in Victoria, Australia, the total distance covered during training was correlated with a higher win rate and with the number of wins and places per start achieved in the previous season. Moreover, training at gallop speeds of 13.3–14.3 m/s over distances of 5000–12,500 m also correlated with a higher rate of career placements per start [3]. Slower-speed gallops (<13.3–14.3 m/s) ranging between 5000 m and 12,500 m also correlated with a higher rate of career placements per start, with similar results observed by other authors [4]. Higher cumulative high-speed (gallop and race) distances and having raced in the previous 30 days correlated with an increased likelihood of horses winning a race [5]. Furthermore, incorporation of periods of rest (‘spells’ in Australia) is common practice and also associated with greater prizemoney per start acquired in the previous race season [3]. Several studies explored correlations between race performance (i.e., measured by Timeform ratings, placements, wins, prizemoney earned, etc.) and lactate concentration (e.g., VLa4 or V4 and post-exercise lactate concentration) [6], heart rate (e.g., HR during exercise tests, V200) [7] and size [8], haematocrit, uric acid [9] and blood gases [10].

Wearable technologies have now facilitated the measurement of such parameters in the field [11,12,13]. Combining on-horse telemetry to measure heart rate, with global positioning satellite (GPS) data for information on speed and distance suggested that heart rate recovery (i.e., heart rate at one minute after strenuous work) could be used as a predictor of fitness or readiness to race, and perhaps race performance (i.e., finishing in the top three, the ‘podium’) [14]. Elite racehorses reach peak speed by predominantly increasing stride length rather than stride frequency [15]. Over the last 200 m of any race, horses that can maintain stride length and/or frequency tend to achieve a better finishing position [16]. Many of these studies, however, were conducted on relatively small datasets. Investigating a large population of elite athletes, such as racehorses, could provide unique insights into the relationship between physiological variables observed in training and performance on race-day. To date, no study has related race performance over multiple seasons with combined information on stride acceleration and cardiovascular fitness in race-pace training sessions. 

Hence, in the present study, a retrospective, observational study was conducted using a fitness tracking device to study peak stride (length and frequency), speed (last 200 and 600 m) and heart rate (peak and recovery) in racing. Thoroughbreds are categorised according to their subsequent racing result (win or lose). By categorising racehorses based on speed (slow, medium and fast), we investigated if variations in recovery parameters during training sessions before races could predict race performance within each speed category. The primary hypothesis of this study is that cardiovascular fitness (rate of recovery of heart rate) and/or stride acceleration (the average time in seconds over consecutive 200 m from a standing start) during race-pace workouts (hard gallop or jumpout sessions only) may predict racing performance (win versus not win). Secondary hypotheses were that relatively high daily temperature and/or humidity may significantly blunt racehorse recovery of heart rate and confound prediction of race performance.

## 2. Materials and Methods

### 2.1. Databases

This retrospective, observational study used two large datasets pertaining to the same cohort of racehorses as follows:(1)Training sessions: collected by means of a fitness tracker (the ‘Equimetre’™, Arioneo Ltd., Paris, France) from a single racing yard (Ciaron Maher Racing) in Victoria, Australia.(2)Race results from the same cohort: available upon subscription in Australia, with race data recorded and downloaded from http://www.racing.com (accessed on 16 January 2023).

### 2.2. Horses

The study population represented a convenience sample and included 485 Thoroughbred racehorses that were recruited from the same racing yard in Australia. The horses were aged between 2 and 10 years at the start of training. For n = 10 horses, age was not recorded. The sex of the horse was coded as male (colts/stallions; n = 108), female (fillies/mares; n = 229) and gelding (n = 148). The sex of the horse was randomly checked online at http://www.racing.com (accessed on 3 April 2023). The trainers regarded all horses as race fit; that is, horses that were actively in training to sustain their fitness levels and competing in races during the study period. Inclusion criteria were horses in training recorded during fast gallop, ‘jumpout’ and ‘barrier trial’ sessions on turf only. For prediction of race performance, a horse must have had ≥1 training session, conducted less than 60 days before any race, with available data for race success. Exclusion criteria were sessions at ‘slow’ intensities (slow/medium/hard canter) and any race data that were available for an individual horse but that preceded any available training data.

### 2.3. Equipment

Horses wore their regular tack and were exercised by a randomly assigned work rider, who varied according to individual training sessions. A tracking device (the ‘Equimetre™’, Arioneo, Ltd., Paris, France) was fitted to the girth prior to training by persons accustomed to using the device, as previously described [17]. The device recorded locomotory parameters (peak stride length and frequency) along with speed (by GNSS) and cardiovascular parameters (HR and HR recovery at 1 min after exercise), as previously described in detail [11,15]. Prior to this study, trainers had previously integrated the use of such trackers into the training regimes for each horse.

### 2.4. Training Data

The trainer determined the nature of each individual training session, directing the work rider as appropriate. The Equimetre™ (Arioneo Ltd., France) was not systematically placed on each horse for every individual training session, but rather for specific sessions. Data for all recorded training sessions was downloaded to an MS Excel sheet. Data included: GNSS (GPS + Glonass + Galileo) satellite data giving information on speed (i.e., time taken to cover 200 m in seconds) measured for each 200 m segment (at 200, 400, 600, 800, 1000 and 1200 m) on a turf racetrack. Based on the GPS coordinates of the device, a virtual timed trial of 1000 m towards a known and virtual finish line at 0 m was generated for each race-pace session, of which jockeys were aware. Consequently, a timed trial with timings recorded at each 200 m segment (i.e., one furlong) and then quintiles of the first and third 200 m speeds (slow, slow-medium, medium, medium-fast and fast) were manually formed to categorise horses accordingly. Speed from the third 200 m mark, at 600 m into any effort, which is often where horses reach peak speed, was then taken and used to categorise horses as being relatively ‘fast’ or ‘slow’ over the final three 200 m sections to a virtual finish line at 0 m. 

All training data were collected between 18 January 2021 and 14 January 2023 and comprised a total of 4837 training sessions (hard gallop n = 3418 sessions and jumpout n = 1419 sessions). Horses would train all year-round throughout the racing season, during which environmental temperatures and humidity can vary considerably. The full dataset comprised n = 485 racehorses completing n = 6 (1–38) training sessions, median (first-third interquartile range [IQR]). Race-pace efforts in training were categorised sequentially by date from 1 to 25, with only n = 14 horses completing 25+ training sessions. Therefore, these sessions were grouped into a 25+ category. The mean interval between training sessions using an Equimetre was 29 ± 49 days (mean ± S.D.). Days in training were also recorded for each horse and was 119 (78–168) days (median, IQR). Any horses completing only a single training session prior to a given race (n = 11 of 485) were removed only from the combined (training and racing) dataset for prediction of race performance (n = 474 racehorses, n = 3418 fast gallop training sessions). From the exact date of training, together with the exact race date, the number of days prior to each race plus the interval in days between races was calculated and included in the dataset for each individual horse.

Prior to any fast training session, horses would be ‘warmed up’ over 0.5–1.5 k of mixed trot/canter. The Equimetre directly recorded aspects of each horse’s cardiovascular responses to exercise, such as heart rate (HR), locomotion and distance. Considered parameters include the following: -Heart rate (HR), measured every second.-Average HR during the gallop, averaged across the time-period of the fast gallop.-Peak heart rate (HRpeak), the highest recorded HR for that session.-Heart rate recovery (HRR), difference between the average HR during the gallop and the HR at 1 min after exercise.-Early, mid and late-phase HRR, as for the above HRR, but the difference was measured at 2 min (early), 3 min (mid) and 4 to 15 min after gallop exercise (late-phase).-Speed, measured in kilometres per hour. Recorded continuously using GPS.-Peak stride length (metres). Recorded continuously.-Peak stride frequency (strides per second). Recorded continuously.-deltaHR, absolute value in beats per min for HR gallop minus HR at 1–5 min recovery.-Environmental temperature and humidity were obtained at the start of training from the nearest weather station, using the median of the latitude and longitude recorded.

The final datasets were checked for artefacts and corrected accordingly in MS Excel. 

### 2.5. Racing Data

The same cohort of racehorses (n = 485) participated in a total of 3810 races. In this study, race class was recorded as follows: Group 1 (n = 93 of 3810 races, 2.44%), Group 2 (n = 73 races, 1.92%), Group 3 (n = 143 races, 3.75%) and Listed (n = 182 races, 4.77%), whereas all other races were manually labelled as Uncategorised (n = 3289 races, 86.3%). Race distance was known from http://www.racing.com (accessed on 18 January 2023) and was classified as a ‘Sprint’ race < 1600 m (n = 2246 races, 77%), ‘Miler race’ 1601–2500 m (n = 521 races, 18%) or ‘Stayer’ race’ > 2501 m (n = 138 races, 5%), as previously described [15]. From the finish position, where available (n = 54 had no finish position recorded), the proportion win (n = 608 occurrences) or lose (n = 3148 occurrences) could be calculated. Races conducted on synthetic tracks were excluded from this study. Races occurred between 19 January 2021 and 17 January 2023, for which racetrack conditions were known for each individual race. 

### 2.6. Data Analysis

*Analysis of training data:* Any normally distributed descriptive data are presented as the mean (±1 standard deviation [SD]). Data distribution was checked either by standard tests (e.g., the Shapiro–Wilk test) or by checking residuals post-analysis. If necessary, the data were log-transformed (log10) prior to analysis. For some analyses, where assumptions for analysis of variance (ANOVA) could not be met due to occasional missing data (e.g., artefacts removed or no data present), linear mixed models (restricted maximum likelihood; REML) were used, with the main effect of interest fitted as a fixed effect and HorseID fitted as a random effect. For analyses of sequential speed or stride in the same horse over repeated 200 m segments, a repeated-measures ANOVA or REML with time fitted as a fixed effect was used. To account for multiple training sessions by the same horse, each training session was coded (1–25) by date, and analyses blocked by HorseID.

*Analysis of racing data:* For any aspect of performance in training (e.g., speed or heart rate recovery), the two datasets were combined according to the individual horse, including all instances of gallop training with each race result. Any training data were excluded if the date was ≥60 days before any race. To determine if training data predicted race success, race performance was categorised binomially as win or not, as previously described [15]. Model fit and significant variables included as fixed effects were assessed using backwards stepwise regression; that is, they were included if univariable analysis suggested importance in the model (*p* ≤ 0.10). Using this method, the final logistic regression model was built, including racehorse age, sex, track condition, and race class as fixed effects with win (yes/no) as the outcome. The training variable of interest was always fitted last in any model. All data were analysed using Genstat v23 (VSNi Ltd., Rothamsted, Harpenden, UK). Statistical significance was accepted at *p* < 0.05.

## 3. Results

### 3.1. Descriptive Characteristics of the Racehorse Training Dataset

Hard gallop and jumpout sessions were ‘race-pace’ and speed (~60 km/h), corresponding to previously reported benchmark data (i.e., 10–12 s/200 m or furlong; [11,15]). Heart rate was at its peak (~200–210 bpm), and peak stride length and frequency were similar to previously reported race efforts [15,18]. The main work distance was less in jumpout versus gallop, with all other descriptive measures indicative of racehorses working at maximum capacity (Table 1). 

### 3.2. Racehorse Heart Rate and Recovery during Training

Heart rate measures analysed across the entire population increased, from trot through canter to hard gallop, peaking at ~210 bpm then falling rapidly over the first 5 min after exercise (Figure 1). Such a profile allowed identification of the period of greatest heart rate recovery for each horse and each session—essentially the difference in heart rate during gallop or at peak to the first minute after exercise (Figure 1). Since the difference in HR with age of horse reduced from gallop to peak, that is, all horses peaked at a similar HR, and no observable or significant difference was noted with age of horse (from 2 to 10 years of age), the fall in HR to 1 min after the average achieved during the gallop was considered indicative of the rate of heart rate recovery (HRR). Thereafter, HR decreased linearly to 5 min before declining at a consistent but much reduced rate to ~100 bpm at 15 min post-exercise. 

As a proof of the principle of heart rate measurement in racehorses being indicative of the intensity of the workout, it was clear that while the average during gallop nor the peak were different, the rate of recovery to 1, 2, and 3 min after exercise were reduced after jumpout versus gallop (effect size ~20 bpm at 1 min, Figure 2). Thereafter, values were similar at 5 min but were again higher (albeit non-significantly at 15 and 30 min). Hence, the phases of HRR could be categorised as early, mid and late (Figure 2). The early phase of recovery corresponded to mean differences (gallop versus jumpout) of ~23 bpm at 1 min. The ‘mid-phase’ (first to third minute) corresponded to a mean difference of ~21 bpm at 2 min and ~4 bpm at 3 min (Figure 3). The ‘late phase’ of HRR was designated as the change in HR from 3–15 min post-exercise, when recovery was gradual and slow. At this stage, no overall differences were noted between training intensities. 

Over the duration of the training period, average temperature but not humidity varied throughout the year from minus 1–3 °C up to ≥28 °C (Figure 4). With all training efforts conducted early in the morning (at the time the environmental readings were taken), we could interrogate whether HR recovery or other indices of racehorse health and wellbeing were affected by environmental conditions. Temperature was divided into quintiles from relatively ‘hot’ (≥18 °C; 934 sessions at this temperature) to relatively ‘cold’ (≤5 °C; 954 sessions). Humidity was divided into quintiles from relatively high (high-humidity, ≥95%; ‘wet’) to low (low-humidity, ≤67%; ‘dry’). In addition, we restricted the dataset to gallop, excluding ‘jumpout’ as the distribution of gallop training sessions was more even throughout the training calendar. When accounting for differences in racehorse age, sex, days-in-training and number of training sessions completed (all were added as either fixed effects or co-variates), horse HR fell more steeply after exercise during hotter temperatures, although the effect size was biologically small. For example, at 2 min after exercise, HRR was 124 in hot versus 128 bpm in cold effect size ~4 ± 2 bpm (Figure 5A). No difference was observed with the variation in humidity (Figure 5B).

### 3.3. Racehorse Speed during Training

Racehorse speed was categorised to allow for potential prediction of race success, and at the third 200 m section of the timed trial, when speed is highest, the difference in stride length of ~0.17 m was evident between quintiles (from relatively slow, ~38.2 km/h, to relatively fast, ~41.6 km/h) (Table 2).

The time taken to cover successive furlongs from 1000 m away from and to a virtual finish line at 0 m (i.e., five successive 200 m segments) was then compared according to speed classified at the first 200 m (Figure 6A) or third 200 m (i.e., 600 m; Figure 6B). The differences between fast and slow sessions (and thus racehorses) were significantly more marked during the first 200 m (1000 m away from the virtual finish line), with ‘slow’ horses taking longer to cover a furlong (i.e., ~18.5 s) than faster horses (i.e., ~12.8 s). Thus, in order for speed-in-training to potentially predict whether a given racehorse was more likely to win or come in the top three in a race, the classification of speed at 600 m was used as being more reflective of that training effort. After 600 m, the time taken to cover a furlong was comparable between groups up to reaching the virtual finish line.

### 3.4. Racehorse Recovery and Speed according to the ‘Quality’ of the Racehorse for the Prediction of Race-Day Performance

Since race participation and results were known, we then retrospectively fitted categories of racehorse into those that had ever participated in a Group race versus those that had never (‘yes’ or ‘no’, EVER group raced or not; Figure 7) and applied this to the training dataset. Horses that had previously participated in a Group race recovered faster than horses that had not. The greatest differences between groups were observed in the first 1–2 min, where a difference of ~5.5 and 6.7 bpm were recorded, respectively (Figure 7). Thereafter, the fall in heart rate was similar between groups. Similarly, racehorse speed was faster amongst racehorses that had participated in Group races, with the greatest differences being observed at the start of the exercise (e.g., the time taken to cover the first 200 m was ~14.8 s versus ~15.5 s (Figure 8). 

### 3.5. Racehorse Recovery and Speed for the Prediction of Race-Day Performance

Locomotory and cardiovascular data recorded during training combined with race outcomes were then used to explore whether any aspect significantly predicted the chance of horses winning a race, regardless of race quality (Group, Listed or Unclassified). Colts were more likely to win compared to mares and geldings; similarly, for our dataset, while stayers were a smaller proportion of the total horses, they had a greater chance of winning races (Figure 9). Interestingly, the finish speed (i.e., the average speed over the last 600 m or final three consecutive furlongs) was significantly greater in those horses that won relative to those that lost. However, speed measured only at the 600 m marker was not significantly different (Figure 9). Equally, for horses with a faster heart rate recovery in the early phase to 1 min post-exercise, a greater chance of winning was also noted (Figure 9). Nevertheless, since a few horses tended to win a lot and others not at all, when data were adjusted to account for individual variation in potential to win, colts and stayers remained more likely to win races, but aspects of training data, with the exception of finishing speed, were much less significant. The change in HRR showed no predictive effect on winning races (Figure 10).

## 4. Discussion

This study has directly, and objectively analysed in-training racehorse speed and heart rate recovery for the prediction of race-day performance (categorised as winning versus not winning). The speed of heart rate recovery to one minute after a high-speed training session did predict race performance, but the effect was markedly reduced when the variation between different horses was accounted for. In addition, heart rate recovery appeared to be greater during relatively hot, but not relatively dry, weather conditions. Racehorses that were categorised as being relatively ‘fast’ from 600 to 1000 m of a timed race-effort in training relative to those categorised as relatively slow were also more likely to have won a race. Again, the effect diminished but nevertheless remained when variability by horse was accounted for. This study provides evidence to partially support our primary hypothesis; training data can be used to predict which racehorses will a) go on to participate in a group race and b) be relatively more successful. Nevertheless, the absolute difference in practice is small (~4–6 beats/min; ~1–1.5 s/furlong), such that monitoring may not be of practicable use. Hence, we suggest that aptitude for superior performance is variable for each individual racehorse, relying on unique physiological factors. Furthermore, race-day performance prediction can be equally influenced by intrinsic aspects such as the sex of the horse (colts have a greater chance of winning) and their particular aptitude for race distance (stayers were also more likely to win). 

### 4.1. Descriptive Characteristics of the Racehorse Training Dataset

As racehorses initiate their training, milestones are commonly established by trainers to plan their progression towards racing [19]. The first milestone typically involves the registration of a horse with a specific trainer. A trainer is required to officially inform his relevant racing authority or governing body (such as Racing Victoria) that a horse has initiated training in his stable. The second milestone consists of entering a horse in an official trial or ‘jumpout’. In Australia, it is mandatory for any horse to run in a qualifying trial before being entered in a race. The number of completed jumpouts by any individual horse prior to a race varies from one trainer to another. Some trainers only opt for one jumpout and consequently consider the first race as a ‘fitness test’. In the present study, jumpouts occurred approximately 10 days before a race. Racing is perceived as the ultimate training milestone.

With the advancement of field technologies such as portable fitness trackers, the ability to evaluate racehorse speed, stride and heart rate in an environment similar to official race meetings has become increasingly feasible and reproducible [15,17,19]. When routinely observing horses exercise live on the track, it is still common practice for trainers to manually time the fastest workouts by means of a stopwatch. Usually, they will measure the time taken in seconds for a given horse to cover a furlong (s/f). In Australia, speeds of fast gallop were surveyed among 66 trainers established in Victoria. Results outlined timings of 11–14 s/f and speeds of 47.9–65.5 km/h during high-intensity workouts [2]. These values were consistent with those reported in the present study and are comparable to what has been recorded during race-day performance [20]. Although the race distance was greater (i.e., long distance or ‘staying’ races), peak stride length and frequency in the current study training data were similar to those identified by colleagues [21], that is, 7.42 m stride length and 2.34 strides/s, respectively. Peak heart rate did not further increase with training intensity, as previously observed by us and others [22], and was similar in both types of race-pace efforts (jumpout versus hard gallop). 

### 4.2. Racehorse Heart Rate and Recovery during Training

As horses progress in their training programme, so does their cardiovascular fitness. Cardiovascular fitness refers to the ability of an individual to rapidly recover or drop its HR from peak to baseline after an intense effort. Heart rate measures are known to vary from one horse to another as they rely on unique aerobic capacity influenced by breed, age or training [23]. At sub-maximal exercise intensities, HR has been suggested to be a reliable indicator of fitness and/or performance in Thoroughbreds [24,25]. However, in flat racehorses running at top speeds, heart rates peak at ~200 bpm [26], which we have previously shown to occur as early as hard canter, where speed is only at approximately 75% maximum for each horse, and thus, to an extent, the increase in heart rate and/or peak, is unlikely to be predictive of performance [15,19].

To date, there are a few studies that have used the rate of recovery of heart rate or HRR to define equine fitness [6], as opposed to a considerable body of evidence in human athletes [27,28]. Racehorse HRR is proportionate to the intensity of the workout [11] and was reflected in the current study: after a jumpout, a simulated race effort against other horses, HRR was blunted, and heart rate took much longer to reach resting values when compared to those that had completed a hard gallop session. It is also conceivable that other factors that could be different during a jumpout session as opposed to a hard gallop at home, such as excitement, anticipation, a novel environment or a greater number of other horses to gallop against, could have contributed to the delay in heart rate recovery. While the age of the horse was accounted for in statistical models, older horses could be more familiar with the jumpout routine and environment, thus less affected by the excitement of interval exercise than the younger horses [29]. Overall, heart rate recovered most in the first minute after exercise. Interestingly, the post 1 min HRR measure is a common indicator used in human performance and is applied in many athletic sports [27,30,31]. The post 1 min HRR parameter was integrated into a recent study that analysed HRR for the prediction of performance in a cohort of 20 National Hunt racehorses [14]. 

Marked variations in average temperature were observed throughout the study period, but not humidity. It is widely assumed that aspects of environmental heat and humidity may influence cardiorespiratory responses to exercise and recovery in horses. Indeed, since the World Equestrian Games in Stockholm (1990) and the Olympic Games in Barcelona (1992), heat stress in horses has received significant media attention, along with a surge in studies investigating how to preserve equestrian competitions in challenging environmental conditions [32,33,34,35,36,37]. A recent study by [38] evidenced that training for 14 days in a heated indoor arena contributed to the reduction of thermal strain on elite sport horses, facilitating competing in hot weather. Thus, high-level horses can acclimatise to heat while remaining in training, a crucial factor for their participation in events such as the Olympics. 

Another study evaluated a small cohort of five Thoroughbreds exercised on a treadmill and exposed to varying environmental conditions [39]. In this study, heart rate during and after exercise was significantly higher in a hot and humid controlled setting compared to cooler and less humid conditions [39]. However, the study was performed in a controlled laboratory setting, which may not reflect responses in the field. Indeed, when heat and humidity were categorised into quintiles, we found either the opposite or no effect at all: improved recovery of heart rate in hotter conditions with no effect of humidity. The explanation may lie in the fact that the majority of the cohort participating in our study were Australian-bred horses (approximately 75% of the horses), not imported from overseas, and therefore perhaps more ‘acclimated’ to the vagaries of the environment in Melbourne. Perhaps the trainer subconsciously instructed riders to not ride as hard in particularly hot conditions, although analyses of speeds and other data between groups suggest this not to be the case. In the colder winter months, horses may not be as fit as they are coming into the spring racing season, when temperatures begin to rise. Relative to data from treadmill workouts [39], perhaps the evaporative and conductive effect of air rushing past the horse outdoors allows for greater temperature regulation. Finally, the observed trend could be influenced by a majority of lower class horses racing in winter versus the higher quality horses that are more likely to race in hotter periods of the year when the group and listed races are organised.

### 4.3. Racehorse Speed during Training

Whether a human sprinter, a marathoner or a racehorse speed is the driving force behind athletic performance. Speed, acceleration rate and the ability to hold maximum speed over a certain distance are highly sought after performance characteristics. All play a part in building efficient racehorse locomotion acquired through the implementation of suitable training programmes [40,41]. Speed is the combination of stride frequency and length [11,42]. All play a part in building efficient racehorse locomotion acquired through the implementation of suitable training programmes [40,41]. Speed is the combination of stride frequency and length [11,42]. Horses are known to reach peak speeds, first by increasing their stride frequency up to the pace of a gallop, then by increasing their stride length [15]. Our results converge on this principle, as stride length was significantly different between racehorses categorised according to speed at a timed marker. It is likely that the large variation in speed according to the categorised racehorse (i.e., fast or slow at the 600 m mark) in the first 200 m reflects the difference in how quickly a jockey gets the horse up to speed for the instructed session (by the trainer). For example, a horse doing a middle- or long-distance workout will not be going as fast into the first 200 or last 400 m compared to a ‘sprinter’. Depending on the workout, they may only start to increase speeds ≥200 m into the workout (i.e., 800 m away from the finish line). It is therefore assumed that the classification of speed at 600 m was more pertinent to comparing groups up to the virtual finish line. While different types of track and within types, different surface conditions are known to influence racehorse speed, for example, racehorses are faster on firmer ground [43], in this study, the majority of sessions were conducted on ‘good or good-to-firm’ ground, and track condition was included as a potential confounding factor. 

### 4.4. Racehorse Heart Rate Recovery and Speed for the Prediction of Race-Day Performance

The Australian racing industry is relatively homogeneous (i.e., comprising a vast majority of sprinters), facilitating the extraction of annual racing data for the purpose of large observational studies such as the present study. A few previously published studies have involved hundreds of racehorses and thousands of races to explore any association between physiology in training (speed and HRR) and racing outcomes. A first approach consisted of comparing HRR and speed between groups of horses categorised according to previous participation in a group race or not. Horses that had participated in a Group race recovered better and ran faster than the horses with no subsequent Group race experience. This suggests that these physiological data could potentially be used to determine which horses are more likely to at least participate in high-class races. While age was an important factor included in our analyses, there is no doubt that the Australian racing programme offers more substantial incentives to race younger horses (2- and 3-year-olds) in higher class races. It is relatively easier for young Thoroughbreds to qualify for such ‘black-type’ races, for example, inclusion criteria may at times only require the two-year-old to win a relatively standard-class race beforehand. 

Thus, a second approach was to integrate all racehorse characteristics, including physiological data alluded to above (speed and heart rate recovery) and inherent data on the horse itself (age, sex and days in training), to identify whether any parameter may predict a greater chance of winning a race. Overall, stayers appeared to have a greater chance of winning relative to other categories of racehorse (sprinters and milers), as previously published [15]. This is not surprising considering that stayers were older (4–5 years of age), thus possibly more mature or of better ‘quality’ (premium or elite horses imported from Europe) and experienced at races or were better placed in more suitable races by the trainer. Colts were also more likely to win, as opposed to fillies, mares or geldings. More (1999) drew similar conclusions: performing horses were more likely to be male, have started as a 2-year-old, and have acquired more starts in the preceding 12 months [44].

Clearly, having greater ability to go faster—speed—is known to favour superior racehorse performance [45]. In this study, after accounting for individual racehorse variability, finish speed, more than HRR, demonstrated a trend towards greater race success. This aligns with a treadmill study that conducted exercise testing in flat racehorses, where lactate—a product of muscle metabolism—as opposed to post-exercise HR positively correlated with Timeform rating [6]. However, other studies, albeit in National Hunt horses (i.e., having to also jump fences), also found that HRR to 1 min after exercise was a tangible predictor of subsequent race position, such as finishing in the top three [14].

Aspects of genetics [46] and nutrition [47] also contribute to racehorse success on the track but were not addressed in this study. Furthermore, the influence of the trainer has also been reported to be significantly associated with racing performance [48]. We could not account for the trainer’s entry strategy for each individual horse. Relying on official racing databases may not fully adjust for the health or sub-clinical injury status of the horses. Additionally, the racing dataset did not provide information on the number of horses participating in the race, which could have skewed the data. Another limitation could have been related to the success criteria established for race performance, which could equally influence the results. For example, certain aspects of training were predictive of race performance when classified as ‘win’ or not, but not when classified as ‘podium’ (i.e., finishing in the top three). We also acknowledge that other previously established race success criteria relate to total earnings [49], Timeform rating [50], handicap value [51] or racing time [52]. Furthermore, the effect of the number of races previously raced by each horse was not taken into account when assessing the prediction of a win or not. Because of the limited number of Equimetre fitness trackers used by the training yard, not all data were recorded, and only faster work was captured. It should also be noted that the study is a convenience sample of racehorses that have used an ‘Equimetre’ intermittently, but in a repeatable fashion in training. The study is retrospective and observational. One test of our data would be to conduct the same study using a larger database to confirm or contradict the minimal effect sizes across the different speeds and in those racehorses, either winning races or not. Alternatively, another prospective test of the data could involve establishing benchmarks of speed and heart rate data according to racehorse age in previously identified winning horses. Lastly, as horseracing regulations continue to evolve, these may, in the near or distant future, allow for the authorisation of tracking devices during races. This perspective would allow researchers to further explore racehorse performance, health and safety. Nevertheless, any training metrics that could help predict racehorse performance are likely only to be best achieved through greater standardisation of tests conducted in training; specifically, by conducting standardised exercise tests with consistent jockeys, maintaining a set order, training at consistent times of day, over set distances and potentially at varying speeds, could any metric in training likely accurately predict performance.

## 5. Conclusions

In conclusion, this study demonstrates that in-training racehorse physiological data contributes, to some extent, to performance prediction in racehorses. Of all aspects tested, there was a slight indication that the finish speed was predictive of race performance. It can therefore be deduced that winning a race is about a multitude of aspects. Considering speed and recovery data, along with other aspects (i.e., genetics, preferred ground and distance, psychological state of the horse, training programmes and the jockey–horse partnership), may help trainers identify early signals of top performers or whether to rest horses and ultimately build a legacy of promising horses. Such a hybrid approach using data along with experience may contribute to a better selection of horses in suitable races and improve overall health and welfare on the track.

## Figures and Tables

**Figure 1 animals-14-01342-f001:**
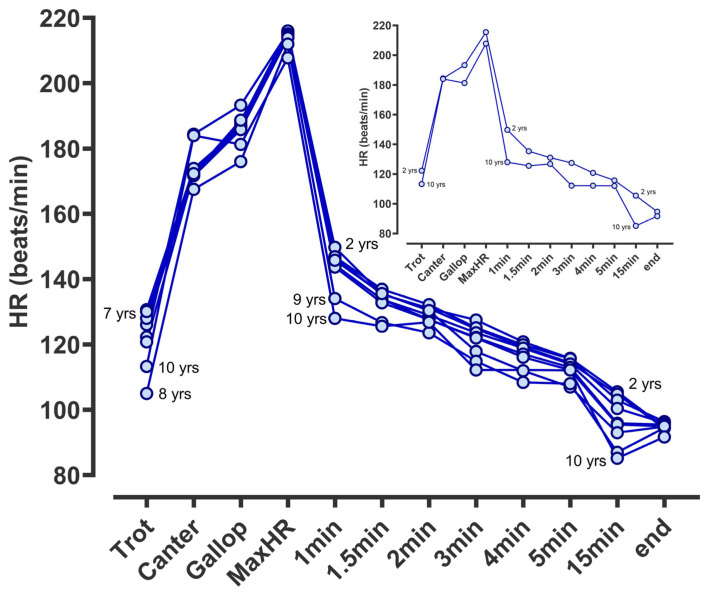
Racehorse heart rate and recovery during hard gallop training according to age. Individual data points are heart rate at varying paces during and after hard gallop training (shown on the x-axis for all horses). Values were recorded by a fitted ‘Equimetre’ on each racehorse (n = 485 different racehorses, n = 4837 different training sessions). Data are predicted means (standard error of the mean, too small to be seen) for each racehorse (adjusted for multiple sessions by the same racehorse), analysed by REML with time as a fixed effect. The age and sex of the horse were included in the statistical model. HorseID and track conditions were included as random effects in the model. Data were analysed using Genstat v23 (VSNi Ltd., Rothampsted, UK). For clarity, the inset graph is the same data but only includes extremes of age.

**Figure 2 animals-14-01342-f002:**
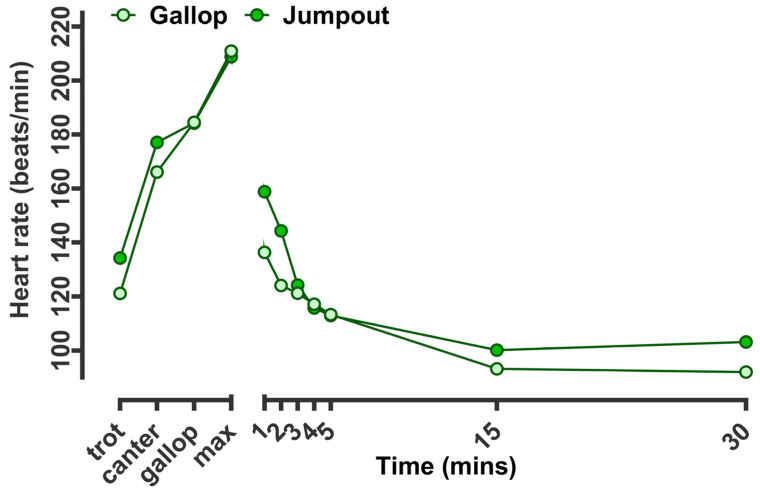
Racehorse heart rate and recovery during training according to exercise group. The data are the predicted mean (±standard error of the mean, too small to be visible) of heart rate for all racehorses (n = 485 different racehorses) in training recorded at gallop (n = 3418 sessions) or jumpout (n = 1419 sessions). Predicted means were calculated from continuous heart rate data recorded for each horse at every minute following exercise, with training type (hard gallop versus jumpout) as a fixed effect in a repeated measures design (time as a fixed effect). HorseID and track conditions were included as random effects in the model.

**Figure 3 animals-14-01342-f003:**
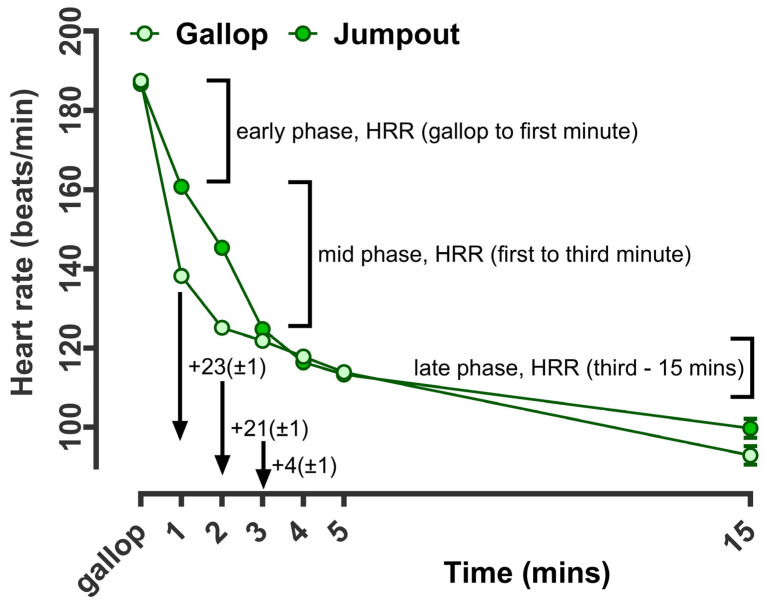
Phases of heart rate recovery during training according to exercise intensity. The data are the predicted mean (±standard error of the mean, too small to be visible) of heart rate for all racehorses (n = 485 different racehorses) in training recorded at gallop (n = 3418 sessions) or jumpout (n = 1419 sessions). Predicted means were calculated from continuous heart rate data recorded for each horse at every minute following exercise, with training type (hard gallop versus jumpout) as the fixed effect in a repeated measures design (time as the fixed effect). HorseID and track conditions were included as random effects in the model. Early, mid and late-phase HRR were categorised to allow potential prediction of race outcomes according to phases of HRR in a logistic regression model.

**Figure 4 animals-14-01342-f004:**
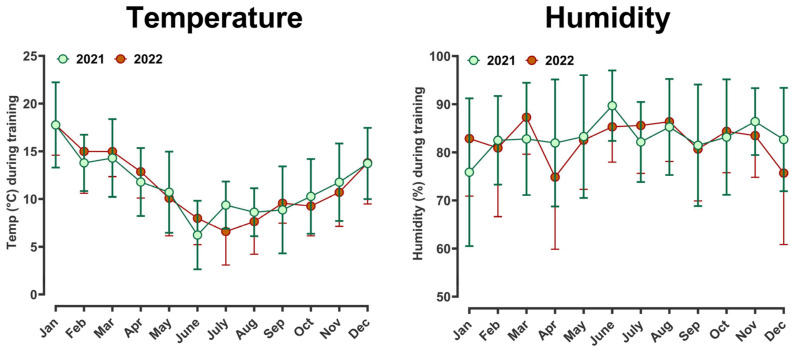
Temperature and humidity according to training month throughout the study period. Data are mean ± standard deviation for observations recorded during every different training session (n = 4837 sessions) as transmitted from the nearest weather station in Victoria, Australia, according to month for the years 2021–2022. In Melbourne, Australia designated ‘Winter’ months with less racing are June, July and August while ‘Summer’ months are December, January and February.

**Figure 5 animals-14-01342-f005:**
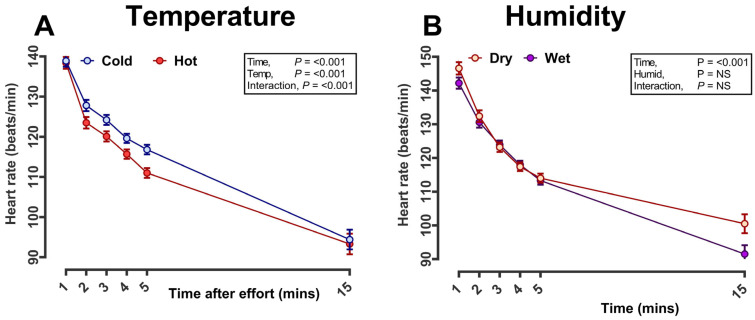
Racehorse heart rate recovery according to environmental conditions during a hard gallop workout: (**A**) temperature and (**B**) humidity. The data are the predicted mean (±standard error of the mean, too small to be visible) of heart rate for all racehorses (n = 485 different racehorses) in training recorded every minute following gallop exercise (n = 3418 sessions). Extremes of temperature were determined from local daily measurements and categorised into quintiles (i.e., first versus fifth quintile of temperature, ‘cold versus hot’) or humidity (i.e., first versus fifth quintile of humidity, ‘dry versus wet’). Each horse had multiple training sessions and was thus included as a random effect in the model. Age, sex, number of training sessions, and days in training were included as potential confounding factors, and means were plotted after adjusting for these effects.

**Figure 6 animals-14-01342-f006:**
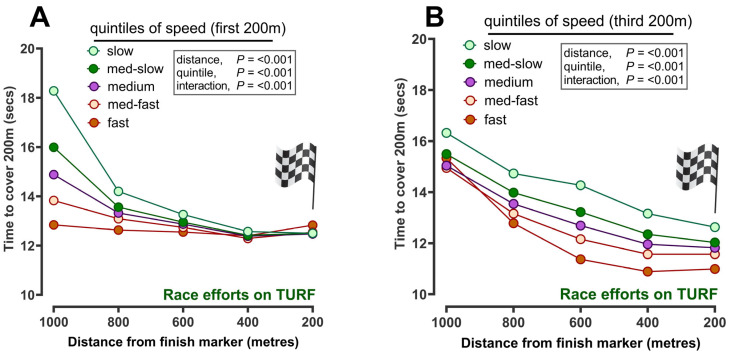
Time to cover sequential 200 m segments to a virtual finish line during gallop workouts: (**A**) racehorses categorised according to speed at first 200 m (‘slow’ to ‘fast’) and then applied to all other times for that horse versus (**B**) racehorses categorised according to speed at third 200 m (‘slow’ to ‘fast’) and then applied to all other times for that horse. The data are the predicted mean (±standard error of the mean, too small to be visible) of time to cover sequential 200 m segments for all racehorses (n = 485 different racehorses) recorded at gallop (n = 3418 sessions). The x-axis describes successive 200 m segments from 1000 m away to a virtual finish line at 0 m.

**Figure 7 animals-14-01342-f007:**
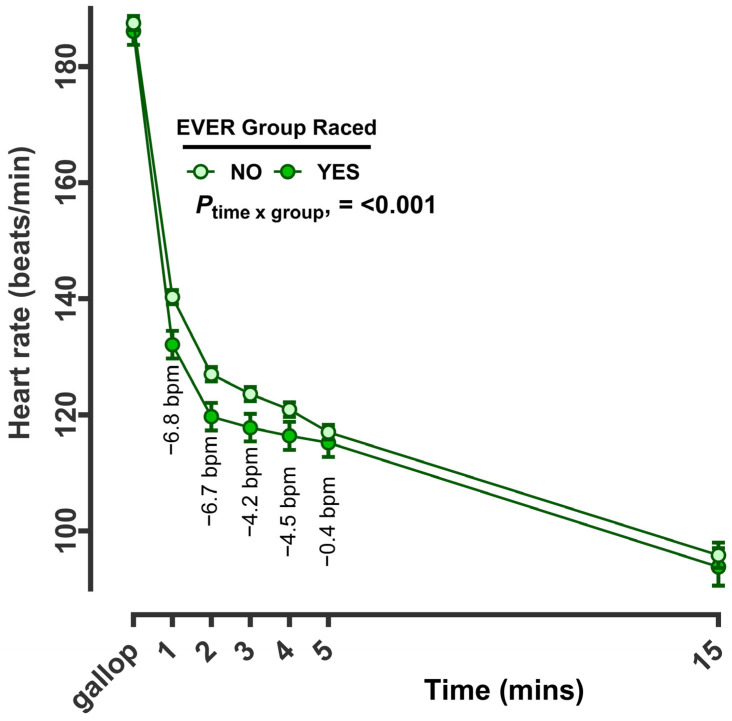
Recovery of racehorse heart rate at, and immediately after, a hard gallop workout, categorised by participation in a group race in their racing career. The data are the predicted mean (±standard error of the mean) for heart rate for all racehorses in training recorded from all gallop sessions. Predicted means were calculated from continuous heart rate data at every minute following exercise with a measured datapoint within range (e.g., artefacts removed). The categorisation of group race participation (yes versus no) was determined from known race data for each individual horse and included Listed, Group 1, 2 and 3 races. Data were analysed by REML in a repeat measures design, with HorseID as a random effect. Age, sex, number of races, days in training and track conditions were included as co-variates. Numbers at each min are the average effect size for the difference in HRR. bpm, beats per minute. From a total of n = 4837 training sessions and n = 474 racehorses, n = 244 sessions were completed by a total of n = 20 different horses that had EVER competed in a group race.

**Figure 8 animals-14-01342-f008:**
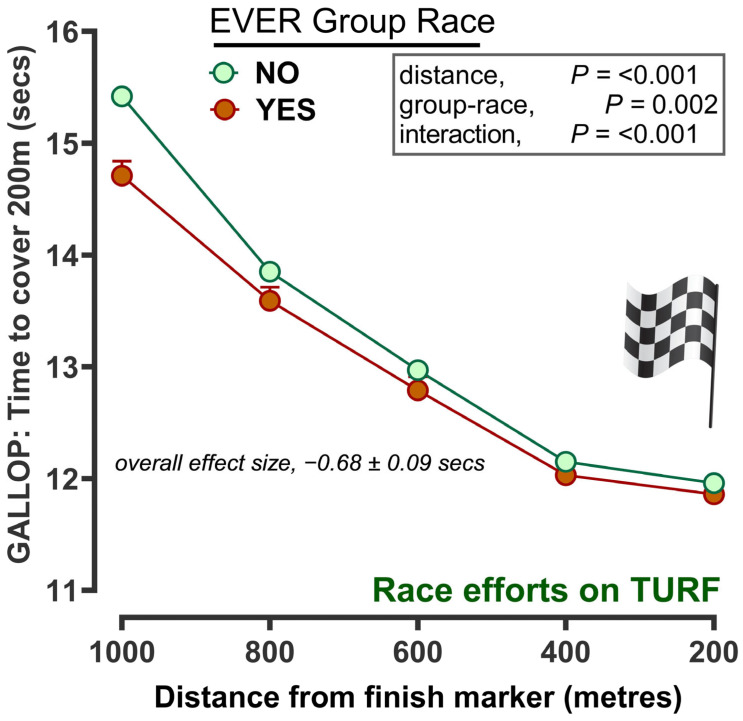
Speed (time to cover a sequential 200 m) during a hard gallop workout according to participation in a group race for each horse in their career. The data are the predicted mean ± standard error of the mean as recorded by ‘Equimetre’. Successive times to cover 200 m from 1000 m to a virtual finish line at 0 m are presented. Participation in a group race at all in their racing career was categorised as ‘yes’ versus ‘no’ and included as a fixed effect in a repeated measures REML analysis. Age, sex, number of races and track condition were included as co-variates in the model. From a total of n = 4837 training sessions and n = 474 racehorses, n = 244 sessions were completed by a total of n = 20 different horses that had EVER competed in a group race.

**Figure 9 animals-14-01342-f009:**
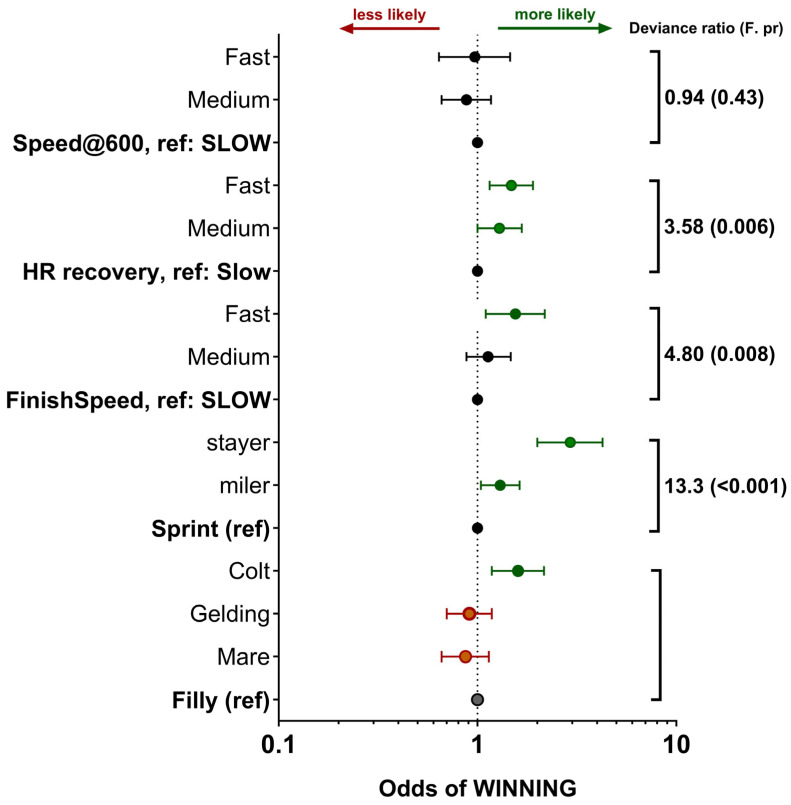
Can training data predict the chance of winning any race. Data are odds ratios ± 95% CI for predicting winning a race from training data. Odds ratios were obtained by integrating known race outcome data from Racing.com (Victoria, Australia; n = 474 different racehorses, n = 3810 different races) with training data (only fast gallop and jumpout on turf) for the same horses before and during two race seasons (n = 3418 different training sessions). Data were analysed with race outcome (e.g., win) as the variable of interest, fitting sex of horse (filly as referent category), racehorse profile (sprinter as referent category), finish speed (slow as referent category), heart rate recovery (slow as referent category) and speed at the 600 m mark (slow as referent category) as other factors of interest. Statistics for main effects are indicated on the right side of the graph with the associated Wald statistic. Statistical significance was accepted at *p* < 0.05.

**Figure 10 animals-14-01342-f010:**
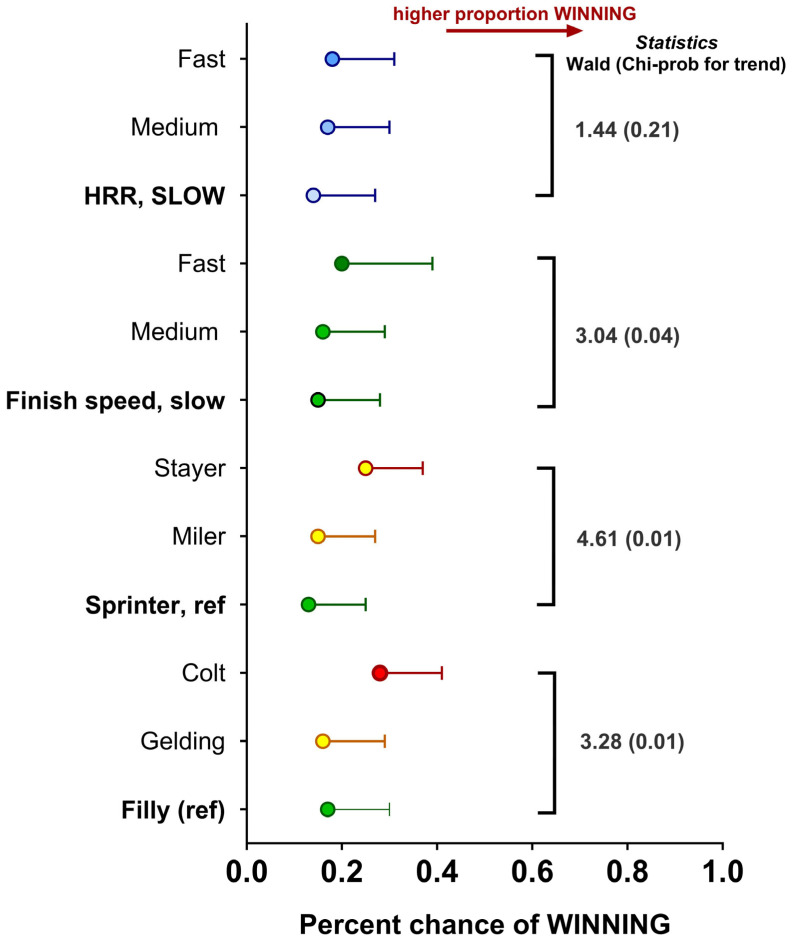
Predicting the percentage chance of race wins on training data. Data are predicted mean log odds (as a percentage) ± SEM for winning a race from training data. Data obtained by integrating known race outcome data from Racing.com (Victoria, Australia; n = 474 different racehorses, n = 3810 different races) with training data (only fast gallop and jumpout on turf) for the same horses before and during two race seasons (n = 3418 different training sessions). Data were analysed with race outcome (e.g., win) as the variable of interest, fitting sex of horse (filly as referent category), racehorse profile (sprinter as referent category), finish speed (slow as referent category), heart rate recovery (slow as referent category) and speed at the 600 m mark (slow as referent category) as other factors of interest. Since the same horse completed multiple training sessions and multiple races, HorseID was fitted as a random effect using Generalised Linear Mixed Models (GzLMM; Genstat v23, Rothampsted, UK). Statistics for main effects are indicated on the right side of the graph with the associated Wald statistic. Statistical significance was accepted at *p* < 0.05.

**Table 1 animals-14-01342-t001:** Descriptive characteristics of the racehorse training dataset.

Training Type	Hard Gallop	Jumpout
Main work distance (metres)	2027 ± 390	1500 ± 254
Max speed (kph)	62.7 ± 3.0	64.3 ± 2.3
Best 200 m (s)	11.6 ± 0.6	11.3 ± 0.4
Best 600 m (s)	36.6 ± 1.9	34.9 ± 1.6
Time last 600–400 m (s)	13.1 ± 0.8	12.0 ± 1.1
Time last 400–200 m (s)	12.1 ± 0.8	11.9 ± 1.4
Time last 200–0 m (s)	11.8 ± 0.9	12.2 ± 1.8
Peak stride frequency (strides/s)	2.39 ± 0.9	2.42 ± 0.09
Peak stride length (metres)	7.35 ± 0.32	7.46 ± 0.29
Peak heart rate exercise (bpm)	212 ± 19	211 ± 17

Values are mean ± standard deviation for continuous data recorded by ‘Equimetre’ (n = 485 different racehorses). A total of n = 4837 different training sessions were recorded and analysed to only include n = 3418 gallop and n = 1419 jumpout sessions for standardisation purposes. All exercise sessions occurred on turf tracks with varying ground conditions. Data were available throughout the year. Time last ‘metres’ was calculated from a virtual finish line based on GPS coordinates and refers to the average time in seconds over consecutive 200 m (i.e., one furlong). ‘Best 200 m, 600’ are the fastest times recorded over those distances for any session for each individual horse. All data are described and not statistically analysed.

**Table 2 animals-14-01342-t002:** Stride length and frequency at 600 m stratified by quintiles of speed. Values are mean ± 1 SD for continuous data recorded by the Equimetre in Australia (n = 485 different racehorses, n = 1419 different gallop sessions). The speed category (slow, medium-slow, fast, medium-fast and fast) was determined based on quintiles of the time taken to cover a furlong at 400–600 m for every individual horse, with a virtual finish line at 1000 m. Data were available throughout the year. Data were analysed by restricted maximal likelihood (REML) for the main effect of the speed category, the individual racehorse as a random effect and after adjusting for significant co-variates (track condition and training session). ^a, b, c, d^ Values within a row with differing superscripts are significantly different at *p* < 0.05, with Bonferroni correction for multiple testing.

Training Type	Slow	Med-Slow	Medium	Med-Fast	Fast	*p*-Value
Peak stride length (m)	7.25 ± 0.02 ^a^	7.30 ± 0.02 ^b^	7.38 ± 0.02 ^c^	7.34 ± 0.02 ^bc^	7.42 ± 0.02 ^d^	<0.001
Peak stride frequency (stride/s)	2.37 ± 0.01	2.37 ± 0.01	2.38 ± 0.01	2.37 ± 0.01	2.37 ± 0.01	0.16
Gallop speed (km/h)	38.2 ± 4.87 ^a^	39.2 ± 4.41 ^b^	39.5 ± 4.41 ^b^	40.5 ± 4.12 ^c^	41.6 ± 4.07 ^d^	<0.001

## Data Availability

All data were collected by Arioneo Ltd. The race results are publicly available (http://www.racing.com; accessed on 18 January 2023), while the training data were previously collected as part of routine recording by the racing yard, which shared the data with the external company (Arioneo Ltd.) that manufactures the data logging device (the ‘Equimetre’). Anonymised training data are available at http://www.arioneo.com.

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
