# Peer review of "Cardiovascular Fitness and Stride Acceleration in Race-Pace Workouts for the Prediction of Performance in Thoroughbreds"

_animals, 2024, doi:10.3390/ani14091342_

Round 1

Reviewer 1 Report

Comments and Suggestions for Authors

This paper looked at leveraging in-training racehorse physiological data to predict race performance, highlighting factors such as colts, stayers, and fast finish speed as significant predictors, potentially aiding trainers in identifying promising horses and improving decision-making.

The introduction outlines the subject well, and introduces the concepts to the reader. The main and secondary objectives are also clearly stated. Materials and methods are well described and are potentially reproducible.

Inclusion and exclusion criteria are missing, even as a convenience sample this should be included.

The results are also adequatly presented. However, the figures need to be bigger as its very difficult to follow without enlarging them, please look into that. The discussion is very well written and organized. Limitations are addressed properly.

References and/ or other mentions in certain brackets show as unreadable, please check the following lines: 277, 291, 311, 313, 338, 350, 351, 352, 374, 384, 385, 404, 408, 411, and many more.

Overall I enjoyed reading this paper and thought it has an interesting approach, and it might have an impact towards the improvement of evidence based selection choices of trainers.

Reviewer 2 Report

Comments and Suggestions for Authors

This is a study with a good potential, including a large cohort of racehorses, which make the data interesting and reliable. However, the paper is written in a very chaotic way, and some paragraphs must be read more than once to be understood. Extensive revision is necessary to further consider this paper for publication.

Here are some general comments:

- Introduction.

The introduction is wide too long (more than 1200 words). It looks like a review, rather than an introduction. Moreover, with being so long, the focus lacks clarity. A more concise introduction (usually, it should be around 500 words) should be written, presenting the essential background to explain the unmet needs of current literature, the reasons for performing the study, and the hypothesis tested.

- Materials and methods.

I understand that this study is based on a very wide dataset, which makes it complex to present in a clear way. However, at present, it is very difficult for the reader to understand the methods.

Regarding included horses, I suggest adding a flowchart or a similar diagram to present the study population: i.e., initial number of horses/trainings included, reasons for exclusion of some horses/training and their number, etc., until the final number of horses/trainings considered in the definitive statistical analyses.

Similarly, I suggest adding a table or bullet points to present in a clear and direct way which parameters were obtained by Equimetre and were consequently included in the statistical analyses.

- Results.

Unfortunately, results are not clear. All graphs must be improved, because at present they do not add anything more than what included in the main text. In all graphs, the authors say that SD/SE are impossibile to see, but I do not think it is acceptable such a justification. Moreover, graph legends are not appropriate, as they should explain what is represented in the graph and help the reader understand it. How analyses were performed, what was included as fixed model, etc. should be explained in the materials and methods, and not in the graph legends of the results section.

Similarly, the paragraph about environmental data has not been anticipated in the materials and methods. The reason for controlling temperature and humidity and how they were evaluated and mitigated in the analyses should be explained in the materials and methods.

In general, the results section should only include the results themselves, and not procedures (which should be included in materials and methods) nor commentaries (which should be included in the discussion).

Moreover, I have noticed in the forest plot that medium and fast heart rate recovery were significantly predicting of winning (p < 0.006), but the abstract says that HR recovery was a not significant parameter. Please, clarify.

Also, supplementary materials including the results of univariable analyses should be provided for transparency.

Finally, all over the text, there are parenthesis containing Chinese ideograms, saying (I have search in Google translate): "Error! Reference source not found". I suppose it is a consequence of using an automatic software for bibliography, which is fine, but at least I expect that the authors revised the manuscript before submitting it. It is difficult not to notice these Chinese ideograms, since there are many throughout the text. 

It is difficult to provide specific comments, as the paper should be extensively revised before evaluating the scientific soundness and revising the discussion and conclusion sections.

Comments on the Quality of English Language

English should be revised. Some sentences are weird and, overall, many paragraphs are difficult to understand.

Reviewer 3 Report

Comments and Suggestions for Authors

The study adds information on the training of Thoroughbred horses in Australia and novel information on the use of technical equipment to collect regularly physiological, locomotor and exercise data. Thank you!

My approach, with the same data, would have been to examine whether the performance of the horses is related to training exercise parameters. Even so, the actual results of the study allow assumptions on the effectivity of training exercise too.

The used approach is valid too. However, it needs to be much better clarified that the exercise used to study the relationships between heart rate, gait as well as speed variables and racing performance of Thoroughbred racehorses was not standardized. Please comment on how likely is it in the training yard studied, that: 1) the trainer asks jockeys to work better horses with more commitment than less good horses? 2) the jockeys themselves engage more knowing that they will exercise a better horse? 3) certain, very experienced jockeys, get to ride the better horses mainly or even only? All of these factors play a role in the race horses world and this holds for the horses in this study too, like you state on page 4, lines 166-167 “The trainer determined the nature of each individual training session, directing the work-rider as appropriate. “

I would add to the conclusions, that trainers could introduce a standardized exercise (test) to increase the likelihood of variables to show a relationship with effectivity of training and maybe even with performance.

Detailed suggestions

Page 1

Line 11 and 12: Change “…whether speed and recovery recorded during fast exercise could…” for “…whether speed during and heart rate recovery after fast exercise could…”

Line 26: Write what means “locomotory profile”

Line 30-31: Explain what means “fast finish speed in exercise” and check conclusion in the simple summary because there is stated, that there was no significant relationship.

Page 2

Line 47; Add “good performing” to “…career longevity of the horses…” because (most of) the other horses do not matter and the boxes are needed for the younger horse crop.

Line 60: Might this be “≥” Instead of “≤“?

Line 67: Add “were” between “…training directly….”

Lines 71 to 78: These lines can be deleted because the information is not related to this study.

Page 3

Line 99: Change “competition” for “competitors”

Page 5

Line 257: Delete “then”

Page 6 to 12: What means 错误!未找到引用源?

Page 9

Question on 3.3. Racehorse speed during training: Please check if the winning horses of all quality categories were exercised faster than those not winning.

Page 11

Line 420: “)” can be deleted

Line 429: “)” can be deleted

Page 13

Line 517: Delete “Takahashi” one time

Round 2

Reviewer 2 Report

Comments and Suggestions for Authors

Dear Authors,

Thank you for your responses to my previous comments and for addressing some of the identified issues. However, there are still several major flaws that have not been addressed and, in my opinion, make the paper not acceptable for publication at present.

Since most of my previous suggestions were not implemented, I will try making a more specific and detailed revision of the introduction and materials and methods sections to make my comments clearer. As there are major limitations in the methods applied, I reserve the right to review the results and discussion only after these deficits have been addressed. Otherwise, the paper is, in my opinion, not enough scientifically sound, transparent, and repeatable to be considered for publication.

I confirm that the topic of the paper is very interesting and the huge amount of data make it potentially highly relevant and significance, since no similar data are reported in the literature. I hoper my comments may help the authors to enhance the scientific quality of the paper, as the data themselves deserve to be published.

Introduction

I appreciate that you shortened the introduction as suggested. However, I still do not find it entirely focused on the background inherent to your study objectives.

-              Lines 52-56: “In Australia, train- 52 ing patterns typically include a high proportion of precision gallop work (e.g. medium 53 gallops at 15s/200m or 13.3–14.3 m/s), with a short sprint (or ‘interval’) at peak speed to- 54 wards the end of the exercise session [2]. Other sessions may be used to simulate race-day 55 experiences such as ‘barrier trials’ or ‘jumpouts’, racing at top-speed against other horses.”

I suggest shortening this part and just presenting the different types of training patterns adopted in Australia, without too many details.

-              Line 58: please remove “form”, too informal. I suggest “...and accumulated fitness.”

-              Lines 58-62: “A prior study in Australia surveyed 66 racehorse 58 trainers in Victoria whom trained their horses at gallop (≥13.3 m/s) for a total of 7,500 - 59 15,000m; total distance correlated with a higher win-rate, previous season wins and places 60 per start [3]. Slower-speed gallops (13.3–14.3 m/s) ranging between 5,000m to 12,500m also 61 correlated with a higher rate of career placings per start.”

Please rephrase. Too many details are not needed for the article’s scope. I suggest something like this: “In a prior study performed in Victoria, Australia, the total distance cover during training correlated with a higher win-rate and with the number of wins and places per start achieved in the previous season. Moreover, training at gallop speeds of 13.3-14.3 m/s over distances of 5,000-12,500 m also correlated with a higher rate of career placings per start.”

-              Lines 62-63: “Similar results were observed by Verheyen, Price [4], Berkman, Teixeira [5].”

This is not a proper way to cite previous study. You could either write “Similar results were observed by other authors”, and report references’ numbers, or report the names of the authors in a correct way. The references are: “Verheyen, K.L., J.S. Price, and J.L. Wood, Exercise during training is associated with racing performance in Thoroughbreds. The Veterinary Journal, 2009. 181(1): p. 43-47.” and “Berkman, C., et al., Distance exercised during submaximal training on race winnings for Thoroughbred racehorses. Ciencia Rural, 695 2015. 45: p. 1268-1273.”. You could either report “Verheyen et al.” or “Verheyen and colleagues”, but it does not make sense to report the surnames of the first two authors only.

-              Lines 63-64: “Another study found that the number of days prior to a race was important in regard to predicting race success [6].”

Please rephrase. Could you clarify what you mean by “number of days prior to a race”? Are you referring to the timing of race-simulation training sessions relative to the actual race day, or is it something else? Please explain. What was predictive of race success?

-              Lines 66-67: “Few studies have correlated physiological parameters with race performance, beyond racehorses having ‘large hearts’ Young and Wood [7]. Similarly, blood lactate concentration sampled after strenuous submaximal treadmill exercise also correlated with Timeform rating and/or subsequent race performance [8].”

Please rephrase. Why are the names of the authors of the reference you cited within the main text without any clear connection? Moreover, I do not agree that FEW studies correlated physiological parameters with race performance. Indeed, SEVERAL studies explored correlations between race performance (measured by Timeform ratings, placings, wins, money earned, etc.) and lactate concentration (e.g. VLa4 or V4, post-exercise lactate concentration), heart rate (e.g. HR during exercise tests, V200), hematocrit, uric acid, blood gas, etc.

-              Lines 80-83: “Equally in human athletes, a positive correlation between stride length and performance has been observed [15]. Similar observations were drawn from a locomotory field- test conducted in equine trotters, using an accelerometer device: maximum speed and stride length were positively correlated with performance index [16]”

Since you already have a lot of data on Thoroughbreds, I suggest focusing the introduction only on this kind of equine athlete, and excluding data from trotters or even humans. Therefore, in my opinion, you should remove this sentence.

-              Lines 93-96: “Using speed to classify racehorses(slow-medium-fast), to observe whether any differences in recovery parameters were apparent in training sessions prior to races, we were further able to determine whether any training parameters within each category of racehorse could predict race performance.”

This sentence is unclear and very difficult to read. Please rephrase.

-              Lines 97-99: “The primary hypothesis of this study is that cardiovascular fitness (rate of recovery of heart rate) or stride acceleration (the average time in seconds over consecutive 200 meters from a standing start) during race-pace workouts (hard gallop or jumpout sessions only) predict performance (win versus not win).”

From the rest of the paper, I understand that rate of HR recovery was not the only parameter considered for HR. Moreover, too many parentheses make the sentence very difficult to read. Therefore, I suggest rephrasing this sentence in a less detailed way, such as: “The primary hypothesis of this study is that cardiovascular fitness and/or stride acceleration during race-pace workouts may predict racing performance.” Similarly, in the following sentence, I suggest adding “may” or “could”, as these are only hypotheses.

Materials and methods

This section is still too chaotic and difficult to read. In my opinion, the authors have not satisfactorily answered to my previous comments on this section. Moreover, I have found some sentences which are in contrast between each other, which make it even more difficult to understand the applied inclusion and exclusion criteria.

-              Line 108: Here you cite Equimetre for the first time, so you should specify the name of the company and country.

-              Lines 115-117: Concerning the ethical approval, here you report number and institution that approved the study. However, in the “Institutional Review Board Statement” you report the same sentence, followed by “Ethical review and approval were waived for this study. As the study is retrospective and observational no experimental protocols were conducted that required ethical oversight”. Please clarify and be consistent throughout the manuscript.

-              Lines 124-129: Inclusion/exclusion criteria are not consistent throughout the study. You should clarify them, since at present it is not clear which criteria were really applied.

For example, in the response to my previous revision, authors write “The final number of 485 was based on Inclusion/exclusion criteria have been added as similarly requested by Reviewer 1, and having sufficient training sessions per horse (at least 5+) to be able to at least ‘predict’ race performance.”, but in the manuscript it is reported that some horses only took part to 1 training session (see Line 169).

Similarly, authors first report “training sessions for individual horses conducted more than 30 days before any race” as an exclusion criterion (Line 127), and then write “Any training data were excluded if the date was ≥ 60 days (i.e., two months) before any race” (Lines 224-225). So, what was the criterion?

-              Lines 134 and 143: When you report Equimetre, Arioneo, please be consistent in the format used (e.g. “Arioneo, Ltd. Paris, France” vs “Arioneo Ltd, France”).

-              Line 153-156; “Speed from the third 200m marker (i.e., at 600m into any effort) was then averaged across the final three 200m sections (e.g., from 600 –800, 800 – 1,000 and 1,000 – 1,200m) and used to form quintiles of the fastest and slowest sessions for any horse, deemed ‘Fast – top quintile’ or ‘Slow – bottom quintile’ in training”.

This sentence is not very clear. Please rephrase it to improve readability.

-              Lines 161-168: I do not understand the reason for inserting here the discussion about the spell period. Please remove it or move it in a more appropriate part of the paper. This is not a method.

-              Lines 168-170: “In this dataset, an average of 6 (1 – 38) training sessions median (first-third interquartile range [IQR]) were recorded per racehorse”.

As above: there are some horses that only had 1 training recorded. But, in the response to reviewers, the authors say that having 5+ trainings recorded was an inclusion criteria. Moreover, please rephrase this sentence: first, authors use “average”, which is a synonym of “mean”, a different concept than median. Also, “training sessions median” without any punctuation is not clear.

-              Lines 174-175: “Any horses completing only a single training session were removed from the dataset.”

As above: only horses completing one single training were excluded, or all those completing less than 5 trainings? Moreover, after their exclusion, how many horses remained and were included in the final analyses? It is not clear whether the number 485 of horses includes also those with only 1 training recorded or not. This is one of the reasons why I suggested including a diagram or flowchart in the horse population paragraph.

-              Lines 168-174: As we are in the materials and methods section, I suggest not including the results of the analyses at this point.

-              Lines 180-188: I find this paragraph not clear. As already suggested in the previous round of revision, my advice is to present the considered parameters (which are not few) in a more schematic way. For example:

“The Equimetre recorded aspects of each horse’s cardiovascular responses to exercise, such as heart rate (HR), locomotion and distance. Considered parameters included:

o   Average HR during gallop

o   HRpeak: peak heart rate

o   HRR: difference between average HR during gallop to HR at 1min after exercise

o   HR at 2, 3, 4, 5 and 15min after exercise

o   DeltaHR: difference between HR during gallop – HR at 1-5 min recovery

o   Speed

o   Stride length

o   Stride frequency

o   Distance.

Moreover, it is not clear, what you considered of speed, stride length and frequency? The average during work, average during each phase of the gallop, the maximum, etc. Please clarify.

Finally, it is not  clear to me the difference between HRR and deltaHR. The authors define HRR as “difference between average HR during gallop to HR at 1min after exercise” and deltaHR as “HR gallop – HR at 1-5 min recovery”, which look the same to me. Also, what do you mean by “at 1-5 min recovery”? Is it HR at 1 min or 5 min post exercise? Please clarify.

-              Lines 194-196: “In this study, race class was recorded as either a Group, Listed or unclassified race: Race data comprised; Group 1 (n = 93 of 3,810 races; 2.44 %), Group 2 (n = 73 races, 1.92 %), Group 3 (n = 143 races, 3.75 %) and Listed (n = 182 races, 4.77 %), whereas all other races were manually labelled as Uncategorised (n = 3,289 races; 86.3%).”

Please rephrase. The punctuation is awkward, and the authors repeat the same concept twice, just adding the number of races included in each category. I suggest rephrasing as follows:
“In this study, race class was recorded as follows: Group 1 (n = 93 of 3,810 races; 2.44 %), Group 2 (n = 73 races, 1.92 %), Group 3 (n = 143 races, 3.75 %) and Listed (n = 182 races, 4.77 %), whereas all other races were manually labelled as Uncategorised (n = 3,289 races; 86.3%).”

-              Lines 205-206: “Other aspects of the dataset such as venue, track condition, carried weight, handicap, rating and prize money were recorded.”

Were these data used throughout the study? Were they included in the statistical analyses? Please specify. If they were not used, they should not be included in the methods. In the statistical analysis section, I only see “racetrack condition” included in the final logistic regression model. Were the other variables included in the univariable models, and excluded from the multivariable model for not reaching significance? Please specify.

-              Lines 209-210: the content of the parenthesis “(e.g., peak stride length, stride frequency)” may be removed. You may whether specify the distribution of all parameters, or of none of them.

-              Lines 219-220: “To account for multiple training sessions in the same horse then training session was coded (1 – 25) by date, and analyses blocked by HorseID.”

If the training sessions were coded from 1 to 25 only, what about the horses classified as 25+ (see Line 171-172)? How were the exceeding training sessions considered?

-              Line 222-223: As above, please remove the parenthesis. You could either specify all the parameters or none of them. Citing two parameters as an example does not add any value to the sentence in my opinion.

-              Lines 224-225: “Any training data were excluded if the date was ≥ 60 days (i.e., two months) before any race.”

See the comment above. Among the exclusion criteria, it is reported “training sessions for individual horses conducted more than 30 days before any race”. Was it 30 or 60 days?

-              Lines 227-229: “Model fit and significant variables included as fixed effects were assessed using backwards stepwise regression; that is, they were included if univariable analysis suggested importance in the model (P ≤ 0.10)”.

In the previous round of revisions, I suggested including the results of the univariable analyses as supplementary materials. Although only the final multivariable model is of interest and deserves to be included in the main manuscript file, it would be advisable to offer readers access to the univariable results. This would enable them to review all parameters considered (such as weight, prize money, etc., as mentioned in previous comments), follow the statistical process, and understand why they were excluded from the final model.

Comments on the Quality of English Language

English quality has improved, but there are still some awkward sentences that need to be rephrased.

Author Response

Dear Authors,

Thank you for your responses to my previous comments and for addressing some of the identified issues. However, there are still several major flaws that have not been addressed and, in my opinion, make the paper not acceptable for publication at present.

Since most of my previous suggestions were not implemented, I will try making a more specific and detailed revision of the introduction and materials and methods sections to make my comments clearer. As there are major limitations in the methods applied, I reserve the right to review the results and discussion only after these deficits have been addressed. Otherwise, the paper is, in my opinion, not enough scientifically sound, transparent, and repeatable to be considered for publication.

I confirm that the topic of the paper is very interesting and the huge amount of data make it potentially highly relevant and significance, since no similar data are reported in the literature. I hope my comments may help the authors to enhance the scientific quality of the paper, as the data themselves deserve to be published.

Thanks very much for your time on this as a reviewer, which I appreciate does not get sufficiently rewarded. We can put an acknowledgment in said section. The data are interesting, and tbh we can make available in an anonymized form if required.

Introduction

I appreciate that you shortened the introduction as suggested. However, I still do not find it entirely focused on the background inherent to your study objectives.

-Lines 52-56: “In Australia, training patterns typically include a high proportion of precision gallop work (e.g. medium gallops at 15s/200m or 13.3–14.3 m/s), with a short sprint (or ‘interval’) at peak speed towards the end of the exercise session Other sessions may be used to simulate race-day experiences such as ‘barrier trials’ or ‘jumpouts’, racing at top-speed against other horses.”
I suggest shortening this part and just presenting the different types of training patterns adopted in Australia, without too many details.

Shortened to: In Australia, racehorse training typically comprises routine canters (<13.3m/s), gallops (13.3–14.3 m/s), and pre-trial gallops (≥13.3 m/s), coupled to alternative methods including mechanical walkers, treadmills, and swimming [2].

-Line 58: please remove “form”, too informal. I suggest “...and accumulated fitness.”

Removed.

-Lines 58-62: “A prior study in Australia surveyed 66 racehorse 58 trainers in Victoria whom trained their horses at gallop (≥13.3 m/s) for a total of 7,500 - 59 15,000m; total distance correlated with a higher win-rate, previous season wins and places 60 per start [3]. Slower-speed gallops (13.3–14.3 m/s) ranging between 5,000m to 12,500m also 61 correlated with a higher rate of career placings per start.”

Please rephrase. Too many details are not needed for the article’s scope. I suggest something like this: “In a prior study performed in Victoria, Australia, the total distance cover during training correlated with a higher win-rate and with the number of wins and places per start achieved in the previous season. Moreover, training at gallop speeds of 13.3-14.3 m/s over distances of 5,000-12,500 m also correlated with a higher rate of career placings per start.”

Fine, sentence rephrased.

-Lines 62-63: “Similar results were observed by Verheyen, Price [4], Berkman, Teixeira [5].”
This is not a proper way to cite previous study. You could either write “Similar results were observed by other authors”, and report references’ numbers, or report the names of the authors in a correct way.

The references are: “Verheyen, K.L., J.S. Price, and J.L. Wood, Exercise during training is associated with racing performance in Thoroughbreds. The Veterinary Journal, 2009. 181(1): p. 43-47.” and “Berkman, C., et al., Distance exercised during submaximal training on race winnings for Thoroughbred racehorses. Ciencia Rural, 695 2015. 45: p. 1268-1273.”. You could either report “Verheyen et al.” or “Verheyen and colleagues”, but it does not make sense to report the surnames of the first two authors only. Reference formatting corrected. Again, we believe this is a formatting issue encountered during submission.

-Lines 63-64: “Another study found that the number of days prior to a race was important in regard to predicting race success [6].”

Please rephrase. Could you clarify what you mean by “number of days prior to a race”? Are you referring to the timing of race-simulation training sessions relative to the actual race day, or is it something else? Please explain. What was predictive of race success?

Sentence rephrased to: ‘While higher cumulative high-speed (gallop and race) distances and having raced in the previous 30 days correlated with an increased likelihood of horses winning a race.’

-Lines 66-67: “Few studies have correlated physiological parameters with race performance, beyond racehorses having ‘large hearts’ Young and Wood [7]. Similarly, blood lactate concentration sampled after strenuous submaximal treadmill exercise also correlated with Timeform rating and/or subsequent race performance [8].”

Please rephrase. Why are the names of the authors of the reference you cited within the main text without any clear connection? Moreover, I do not agree that FEW studies correlated physiological parameters with race performance.

Indeed, SEVERAL studies explored correlations between race performance (measured by Timeform ratings, placings, wins, money earned, etc.) and lactate concentration (e.g. VLa4 or V4, post-exercise lactate concentration), heart rate (e.g. HR during exercise tests, V200), hematocrit, uric acid, blood gas, etc.

Sentence altered to include the suggested points.

-Lines 80-83: “Equally in human athletes, a positive correlation between stride length and performance has been observed [15]. Similar observations were drawn from a locomotory field- test conducted in equine trotters, using an accelerometer device: maximum speed and stride length were positively correlated with performance index [16]”

Since you already have a lot of data on Thoroughbreds, I suggest focusing the introduction only on this kind of equine athlete, and excluding data from trotters or even humans. Therefore, in my opinion, you should remove this sentence.

This is fine, sentence removed.

-Lines 93-96: “Using speed to classify racehorses(slow-medium-fast), to observe whether any differences in recovery parameters were apparent in training sessions prior to races, we were further able to determine whether any training parameters within each category of racehorse could predict race performance.”
This sentence is unclear and very difficult to read. Please rephrase.

Sentence rephrased to: By categorising racehorses based on speed (slow, medium, fast), we investigated if variations in recovery parameters during training sessions before races could predict race performance within each speed category.

-Lines 97-99: “The primary hypothesis of this study is that cardiovascular fitness (rate of recovery of heart rate) or stride acceleration (the average time in seconds over consecutive 200 meters from a standing start) during race-pace workouts (hard gallop or jumpout sessions only) predict performance (win versus not win).”

From the rest of the paper, I understand that rate of HR recovery was not the only parameter considered for HR. Moreover, too many parentheses make the sentence very difficult to read. Therefore, I suggest rephrasing this sentence in a less detailed way, such as: “The primary hypothesis of this study is that cardiovascular fitness and/or stride acceleration during race-pace workouts may predict racing performance.” Similarly, in the following sentence, I suggest adding “may” or “could”, as these are only hypotheses.

Sentences rephrased as suggested.

Materials and methods

This section is still too chaotic and difficult to read. In my opinion, the authors have not satisfactorily answered my previous comments on this section. Moreover, I have found some sentences which are in contrast between each other, which make it even more difficult to understand the applied inclusion and exclusion criteria.

-Line 108: Here you cite Equimetre for the first time, so you should specify the name of the company and country.

Added.

-Lines 115-117: Concerning the ethical approval, here you report number and institution that approved the study. However, in the “Institutional Review Board Statement” you report the same sentence, followed by “Ethical review and approval were waived for this study. As the study is retrospective and observational no experimental protocols were conducted that required ethical oversight”. Please clarify and be consistent throughout the manuscript.

As per a previously published paper in Animals, sentence removed from methods to be mentioned only in Institutional Review Board Statement section (line 656).

-Lines 124-129: Inclusion/exclusion criteria are not consistent throughout the study. You should clarify them, since at present it is not clear which criteria were really applied. For example, in the response to my previous revision, authors write “The final number of 485 was based on Inclusion/exclusion criteria have been added as similarly requested by Reviewer 1, and having sufficient training sessions per horse (at least 5+) to be able to at least ‘predict’ race performance.”, but in the manuscript it is reported that some horses only took part to 1 training session (see Line 169).

We were originally provided with a full dataset of training sessions as recorded by the Equimetre over a given time-period (two years, n=5635 sessions). We also received a separate dataset of race results for that yard (n=3834 races). We first refined the dataset to only those hard-gallop and jumpout sessions for horses that had at least raced. We then considered that training sessions must obviously come before race results, so if we had a race result but no training sessions before, then that race result was removed etc… The two datasets were then combined, with the 485 horses being the common denominator. The descriptives in the paper describe this dataset, with summary measures being a median of n=3 training session per horse and the first-third quartile as (1 – 38). Where you are correct in identifying this error, for any predictive analysis by logistic regression, the instances of n=1 were removed. It was of interest if a horse participated in, say, 1 group race and did well and so for descriptives those instances remained. But, for log regression, then removal of 11 horses with only n=1 race, left us with n=474 horses, n=3418 fast gallop training sessions. This has been updated at appropriate places in the manuscript (e.g. Figure 9, 10 legend).

Lines 162 – 165. Any horses completing only a single training session prior to a given race (n = 11 of 485) were removed only from the combined (training and racing) dataset for prediction of race performance (n = 474 racehorses, n = 3418 fast gallop training sessions).

Similarly, authors first report “training sessions for individual horses conducted more than 30 days before any race” as an exclusion criterion (Line 127), and then write “Any training data were excluded if the date was ≥ 60 days (i.e., two months) before any race” (Lines 224-225). So, what was the criterion?

Sorry, this was an oversight on my behalf. Lines 120 – 121, I have added in For prediction of race performance, a horse must have ≥ 1 training session, conducted less than 60 days before any race with available data for race success.

-Lines 134 and 143: When you report Equimetre, Arioneo, please be consistent in the format used (e.g. “Arioneo, Ltd. Paris, France” vs “Arioneo Ltd, France”).

Updated.

-Line 153-156; “Speed from the third 200m marker (i.e., at 600m into any effort) was then averaged across the final three 200m sections (e.g., from 600 –800, 800 – 1,000 and 1,000 – 1,200m) and used to form quintiles of the fastest and slowest sessions for any horse, deemed ‘Fast – top quintile’ or ‘Slow – bottom quintile’ in training”.

This sentence is not very clear. Please rephrase it to improve readability.

Rephrased to: Speed from the third 200m mark, at 600m into any effort – which is often where horses reach peak speed - was then taken and used to categorise horses as being relatively ‘fast’ or ‘slow’ over the final three 200m sections to a virtual finish line at 0m.

- Lines 161-168: I do not understand the reason for inserting here the discussion about the spell period. Please remove it or move it in a more appropriate part of the paper. This is not a method.

OK, have removed reference to the ‘spell’ period.

- Lines 168-170: “In this dataset, an average of 6 (1 – 38) training sessions median (first-third interquartile range [IQR]) were recorded per racehorse”. As above: there are some horses that only had 1 training recorded. But, in the response to reviewers, the authors say that having 5+ trainings recorded was an inclusion criteria.

Sorry, this was unclear. As referred to above, is should be clearer now. n=1 session in training, was included for descriptive data, but not for any supposed prediction of race performance (thus n=11 horses removed for prediction race performance).

Moreover, please rephrase this sentence: first, authors use “average”, which is a synonym of “mean”, a different concept than median. Also, “training sessions median” without any punctuation is not clear.

Rephrased to: In this dataset, each horse completed n = 6 (1 – 38) training sessions, median (first-third interquartile range [IQR]).

- Lines 174-175: “Any horses completing only a single training session were removed from the dataset.”
As above: only horses completing one single training were excluded, or all those completing less than 5 trainings? Moreover, after their exclusion, how many horses remained and were included in the final analyses? It is not clear whether the number 485 of horses includes also those with only 1 training recorded or not. This is one of the reasons why I suggested including a diagram or flowchart in the horse population paragraph.

As referred to above, is should be clearer now. n=1 session in training, was included for descriptive data, but not for any supposed prediction of race performance (thus n=11 horses removed for prediction race performance). This was the only instance of removing horses after our inclusion/exclusion criteria

- Lines 168-174: As we are in the materials and methods section, I suggest not including the results of the analyses at this point.

Have altered to…The only data included are the Prior to any fast training session, horses would be ‘warmed up’ over 0.5 – 1.5 k of mixed trot/canter. We think it is important to at least state this as a means of preparing horses

- Lines 180-188: I find this paragraph not clear. As already suggested in the previous round of revision, my advice is to present the considered parameters (which are not few) in a more schematic way. For example:

“The Equimetre recorded aspects of each horse’s cardiovascular responses to exercise, such as heart rate (HR), locomotion and distance. Considered parameters included:

o   Average HR during gallop

o   HRpeak: peak heart rate

o   HRR: difference between average HR during gallop to HR at 1min after exercise

o   HR at 2, 3, 4, 5 and 15min after exercise

o   DeltaHR: difference between HR during gallop – HR at 1-5 min recovery

o   Speed

o   Stride length

o   Stride frequency

o   Distance.

 OK, this has been altered accordingly.

Moreover, it is not clear, what you considered of speed, stride length and frequency? The average during work, average during each phase of the gallop, the maximum, etc. Please clarify.

These data are reported in Table 1, but have also been clarified here. These values are recorded continuously according to GPS and algorithms incorporating leg movement as validated by the Equimetre, and as described in previous earlier papers using the Equimetre.

Finally, it is not clear to me the difference between HRR and deltaHR.

The authors define HRR as “difference between the average HR during gallop to HR at 1min after exercise” and deltaHR as “HR gallop – HR at 1-5 min recovery”, which look the same to me. Also, what do you mean by “at 1-5 min recovery”? Is it HR at 1 min or 5 min post exercise? Please clarify.

Fine. So, for each individual horse you have a gallop heart rate, lets say 215 bpm and a point measurement for that horse during recovery at say 1min (e.g. 150 bpm), 2min (e.g. 130) etc… to 15min post-exercise. In a repeated-measures analysis, those datapoints for that horse would be represented as is, in order to describe the central tendency for HRR, as shown for example, in Figure 2, Fig 5 etc.. Delta HR is essentially a similar measure but creates a positive value for phases of recovery, so with above data deltaHR at 1min is (215-150) = 65bpm. At 2min, it is either 85bpm or 20bpm depending on which value preceding it you reference it to. This creates subtle differences that enable different analyses – certain horses have good recovery (big number, large fall in HR) versus poor recovery (value stays high, small number, small drop in HR)…categories of recovery can be created for all sessions and horses. It is a subtle difference but we think important. Perhaps this is clearer with above referred to changes made?

-Lines 194-196: “In this study, race class was recorded as either a Group, Listed or unclassified race: Race data comprised; Group 1 (n = 93 of 3,810 races; 2.44 %), Group 2 (n = 73 races, 1.92 %), Group 3 (n = 143 races, 3.75 %) and Listed (n = 182 races, 4.77 %), whereas all other races were manually labelled as Uncategorised (n = 3,289 races; 86.3%).”
Please rephrase. The punctuation is awkward, and the authors repeat the same concept twice, just adding the number of races included in each category. I suggest rephrasing as follows:
“In this study, race class was recorded as follows: Group 1 (n = 93 of 3,810 races; 2.44 %), Group 2 (n = 73 races, 1.92 %), Group 3 (n = 143 races, 3.75 %) and Listed (n = 182 races, 4.77 %), whereas all other races were manually labelled as Uncategorised (n = 3,289 races; 86.3%).”

Rephrased.

-Lines 205-206: “Other aspects of the dataset such as venue, track condition, carried weight, handicap, rating and prize money were recorded.”

Were these data used throughout the study? Were they included in the statistical analyses? Please specify. If they were not used, they should not be included in the methods.

Sentence removed.

New sentence added on line 200: Races occurred between 19 January 2021 and 17 January 2023, for which racetrack condition was known for each individual race.

In the statistical analysis section, I only see “racetrack condition” included in the final logistic regression model. Were the other variables included in the univariable models, and excluded from the multivariable model for not reaching significance? Please specify.

Only track condition was selected for final incorporation as it had reasonable effects on various parameters and so was the only parameter retained consistently.

-Lines 209-210: the content of the parenthesis “(e.g., peak stride length, stride frequency)” may be removed. You may either specify the distribution of all parameters, or of none of them.

We are just providing an example of classically normally-distributed data versus not. We have previously used this format in a previous Animals paper, but on re-reading, I think I agree. Words have been removed.

-Lines 219-220: “To account for multiple training sessions in the same horse then training session was coded (1 – 25) by date, and analyses blocked by HorseID.” If the training sessions were coded from 1 to 25 only, what about the horses classified as 25+ (see Line 171-172)? How were the exceeding training sessions considered?

In the dataset only n=3 horses in total had data using the Equimetre for more than n=25 training sessions. Thus, in creating this category of 25+ training session it allowed us to retain that information for those horses individually to training session 25, and as a group thereafter. Listing training sessions chronologically, allowed us to maybe see any developments in individual horses during the course of training. The few horses after the 25th training session really, as a group, showed no development (highly variable since so few horses) to the max of n=40 training sessions.

-              Line 222-223: As above, please remove the parenthesis. You could either specify all the parameters or none of them. Citing two parameters as an example does not add any value to the sentence in my opinion.

The text has been removed.

-              Lines 224-225: “Any training data were excluded if the date was ≥ 60 days (i.e., two months) before any race.”

See the comment above. Among the exclusion criteria, it is reported “training sessions for individual horses conducted more than 30 days before any race”. Was it 30 or 60 days?

Clarified above. This was our mistake. 60days was the max in the dataset, not 30. We previously thought 30 days might be more predictive, but doing so reduced size of the dataset and thus power. We therefore stuck with 60 days.

-              Lines 227-229: “Model fit and significant variables included as fixed effects were assessed using backwards stepwise regression; that is, they were included if univariable analysis suggested importance in the model (P ≤ 0.10)”.

In the previous round of revisions, I suggested including the results of the univariable analyses as supplementary materials. Although only the final multivariable model is of interest and deserves to be included in the main manuscript file, it would be advisable to offer readers access to the univariable results. This would enable them to review all parameters considered (such as weight, prize money, etc., as mentioned in previous comments), follow the statistical process, and understand why they were excluded from the final model.

To be honest, the included variables were strongly associated and also fairly obvious – age, race-class. Any other effect were either minimal or confounded to some extent and we had to account for this e.g. stayers tended to be older, geldings. We didn’t have full data for weight added, handicapping for each race and the extensive missing data and confounding with non-handicapped races meant we really couldn’t include many of the aspects you suggest. We think, overall and taken together with what we have produced, the main effects//results are pretty clear and really wouldn’t be considerably affected by addition of another table. If the editor would really like that quite large table adding, in addition to the current two tables and ten figures, we can.

Reviewer 3 Report

Comments and Suggestions for Authors

Thank you very much for the additional information in your response to my comments and queries! Very interesting work!

I understand what you address in different sections of the manuscript about the possible effects of the trainers’ decisions on the results of workouts of horses and therefore measurements. Persons with inside knowledge will understand the relevance of the described limitations too. However, for all those lacking the insight in the race horse universe – these are the majority and becoming more –, and for all others too, it has to become clear to the readers, that the likelihood of predicting racing performance may be increased with a standardized exercise test (same jockey, or few jockeys, same time of the day – which is most often met anyway because those horses that will run fast often will run first-time or early-time of the day –, same order to the jockey, run over defined distances, eventually at different speeds). Please consider adding this in the conclusions.

Several running speeds are used to examine whether they allow to predict performance but it is not always clear which one is meant. For example:

1. In the abstract is written “Colts (P < 0.001), stayers (P < 0.001) and being relatively fast over the last 600m of a benchmark test in training (P < 0.008) were all predictive of race performance. Heart rate recovery after exercise (P = 0.21) and speed recorded at 600m of a 1km benchmark test in training (P = 0.94) were not predictive” and in the simple summary it is written “Heart rate recovery after exercise (P = 0.21) and speed recorded at 600m of a 1km benchmark test in training (P = 0.94) were not powerful as race predictors” only. Write the same in both.

2. Page 10, line 334: It is written “the third 200m section of the timed trial when speed is highest…” but in Table 1 is shown that horses galloped fastest (duration was shortest) during the last 200 m of the Hard Gallops monitored (for Jumpouts it was between 400 and 200 m before the finish line). How is this discrepancy for the Hard Gallops explained?

3. What is “Best 200m” in table 1? Is it: a) the shortest time run in any of the 200m sections? b) Else? Please explain.

In keywords the term training should be included.

Add number of horses / observations in Figures 1, 2, 3, 4, 5, 6, 7, 8

Page 13

Line 405: Add “…ly different….” To “…was not significant….”.

Add n in Figure 7, 8.

Page 18

Lines 572-574: Text can be deleted because it is the repetition of lines 570-572.

Page 19

Lines 627-628: Text can be deleted because it is the same as in lines 625-626.

Author Response

Thank you very much for the additional information in your response to my comments and queries! Very interesting work!

I understand what you address in different sections of the manuscript about the possible effects of the trainers’ decisions on the results of workouts of horses and therefore measurements. Persons with inside knowledge will understand the relevance of the described limitations too. However, for all those lacking the insight in the race horse universe – these are the majority and becoming more –, and for all others too, it has to become clear to the readers, that the likelihood of predicting racing performance may be increased with a standardized exercise test (same jockey, or few jockeys, same time of the day – which is most often met anyway because those horses that will run fast often will run first-time or early-time of the day –, same order to the jockey, run over defined distances, eventually at different speeds). Please consider adding this in the conclusions.

Thank you for your suggestion, we agree. We have added it at the end of the discussion just before the conclusions (Lines 632-637)

Several running speeds are used to examine whether they allow to predict performance but it is not always clear which one is meant. For example:

  1. In the abstract is written

 “Colts (P < 0.001), stayers (P < 0.001) and being relatively fast over the last 600m of a benchmark test in training (P < 0.008) were all predictive of race performance. Heart rate recovery after exercise (P = 0.21) and speed recorded at 600m of a 1km benchmark test in training (P = 0.94) were not predictive” and in the simple summary it is written “Heart rate recovery after exercise (P = 0.21) and speed recorded at 600m of a 1km benchmark test in training (P = 0.94) were not powerful as race predictors” only. Write the same in both.

Perhaps this is a misunderstanding of the text, and we are unsure how to make it clearer: they are essentially two different parameters: 1) the ‘last 600m’ or ‘finish speed’ = the average speed over the last 600m or final three consecutive furlongs, whereas 2) speed at the 600m mark = individual data point at the exact point of being 600m into the training test. We feel this is spelled out in the methods, but appreciate it is quite complicated.

  1. Page 10, line 334: It is written “the third 200m section of the timed trial when speed is highest…” but in Table 1 is shown that horses galloped fastest (duration was shortest) during the last 200 m of the Hard Gallops monitored (for Jumpouts it was between 400 and 200 m before the finish line). How is this discrepancy for the Hard Gallops explained?

When averaging all the data as described in the table, then jumpout sessions remain faster than hard gallop (12.03 seconds versus 12.33 seconds). Any difference is small, but we accept that there could be some variation. We cannot objectively describe why this discrepancy exists…it after all only reflects two points on a timed-trial separated by ~100-to-200m or so, and could reflect jockeys, horses slowing slightly, tying up slightly, the jumpout sessions being slightly different to hard gallops in training e.g. trainers, performance analysts standing at the finish line (jumpout).

  1. What is “Best 200m” in table 1? Is it: a) the shortest time run in any of the 200m sections? b) Else? Please explain.

Yes it is exactly a). This has been added to the legend

In keywords the term training should be included.

Added.

Add number of horses / observations in Figures 1, 2, 3, 4, 5, 6, 7, 8

 Added into the legend, Figure 1, 2, 3, 4, 5, 6, 7,8 as it would clutter the graphs.

Page 13

Line 405: Add “…ly different….” To “…was not significant….”.

Added.

Add n in Figure 7, 8.

Yes, agree. Have added and believe it makes the information more transparent.

Page 18

Lines 572-574: Text can be deleted because it is the repetition of lines 570-572.

Removed, thank you.

Page 19

Lines 627-628: Text can be deleted because it is the same as in lines 625-626.

Removed, thank you.